# Melanin-Based Nanoparticles for Lymph Node Tattooing: Experimental, Histopathological and Ultrastructural Study

**DOI:** 10.3390/nano14131149

**Published:** 2024-07-04

**Authors:** Marta Baselga, Antonio Güemes, Cristina Yus, Teresa Alejo, Víctor Sebastián, Dolores Arribas, Gracia Mendoza, Eva Monleón, Manuel Arruebo

**Affiliations:** 1Institute for Health Research Aragon (IIS Aragón), 50009 Zaragoza, Spain; mbaselga@iisaragon.es (M.B.); cyargon@unizar.es (C.Y.); talejo@certest.es (T.A.); victorse@unizar.es (V.S.); mdarribas@salud.aragon.es (D.A.); gmendoza@iisaragon.es (G.M.); emonleon@unizar.es (E.M.); arruebom@unizar.es (M.A.); 2Department of Surgery, University of Zaragoza, 50009 Zaragoza, Spain; 3Instituto de Nanociencia y Materiales de Aragon, CSIC—University of Zaragoza, 50009 Zaragoza, Spain; 4Department of Chemical Engineering, University of Zaragoza, Campus Río Ebro, 50018 Zaragoza, Spain; 5Department of Human Anatomy and Histology, University of Zaragoza, 50009 Zaragoza, Spain; 6Centro de Encefalopatías y Enfermedades Transmisibles Emergentes, University of Zaragoza, 50009 Zaragoza, Spain

**Keywords:** nanoparticles, lymph node, breast cancer, melanin, tattoo, surgery

## Abstract

In breast cancer, Targeted Axillary Dissection (TAD) allows for the selective excision of the sentinel lymph node (SLN) during primary tumor surgery. TAD consists of the resection of labelled SLNs prior to neoadjuvant chemotherapy (NACT). Numerous clinical and preclinical studies have explored the use of carbon-based colloids for SLN tattooing prior to NACT. However, carbon vectors show varying degrees of inflammatory reactions and, in about one fifth of cases, carbon particles migrate via the lymphatic pathway to other nodes, causing the SLN to mismatch the tattooed node. To overcome these limitations, in this study, we explored the use of melanin as a staining endogenous pigment. We synthesized and characterized melanin-loaded polymeric nanoparticles (Mel-NPs) and used them to tattoo lymph nodes in pig animal models given the similarity in the size of the human and pig nodes. Mel-NPs tattooed lymph nodes showed high identification rates, reaching 83.3% positive identification 16 weeks after tattooing. We did not observe any reduction in the identification as time increased, implying that the colloid is stable in the lymph node tissue. In addition, we performed histological and ultrastructural studies to characterize the biological behavior of the tag. We observed foreign-body-like granulomatous inflammatory responses associated with Mel-NPs, characterized by the formation of multinucleated giant cells. In addition, electron microscopy studies showed that uptake is mainly performed by macrophages, and that macrophages undergo cellular damage associated with particle uptake.

## 1. Introduction

Metastasis is a complex process where cancer cells spread from a primary tumor to other parts of the body, forming secondary tumors in distant organs or tissues [1]. In breast cancer, the most frequent metastatic spread occurs in regional lymph nodes, especially in the homolateral axilla. Metastatic invasion in axillary nodes is one of the most clinically relevant prognostic indicators as it is closely related to the increased likelihood of recurrence and mortality, as well as having a direct influence on the treatment strategy to be followed. According to the American Cancer Society, it is estimated that at the time of diagnosis 27% of patients have regional invasion and 6% have distant metastases [2]. Therefore, histopathological analysis of the sentinel lymph node in breast cancer is indispensable for assessing treatment strategies.

The current gold standard to avoid axillary lymphadenectomy (AL) and its significant sequelae is selective sentinel lymph node biopsy (SLNB), since if the sentinel node is not cancer-positive, the likelihood of another axillary lymph node being involved is remote. Therefore, SLNB is recommended to determine the axillary node status [3,4,5,6]. Typically, the technique for SLNB consists of the use of radiotracers (albumin nanocolloids, tin colloids or rhenium sulphide) for preoperative lymphatic mapping and, optionally, in conjunction with the dye-labeling technique or intraoperative probes or imaging techniques, especially if there is difficulty in visualizing the lymphoscintigraphy. Radiotracers are preferably injected intra- or peritumorally, especially in non-palpable lesions, although periareolar or subdermal inoculation is also acceptable. During surgery, when dyes are used, priority is given to the use of isosulfan blue, patent blue V, methylene blue or indocyanine green, although the latter requires cameras with special detectors to macroscopically visualize it. The dye migrates to the SLN, marking the node with an intense blue, facilitating its identification from other axillary tissues [7].

In a large proportion of breast cancer cases, patients receive neoadjuvant chemotherapy (NATC), which is an impediment to performing SLNB. Detection of the SLN in patients who have undergone NATC is a challenge, as the treatment may alter the lymphatic drainage pathways and reduce the size of the SLN. In these cases, the standard for SLNB has high false-negative rates, which may leave a chemoresistant tumor in the axilla and underestimate the lesion [8,9,10,11]. To avoid these limitations, the technique of Targeted Axillary Dissection (TAD) has been established, where SLN marking is performed prior to NATC Among the best known TAD techniques are surgical clips, magnetic seeds, radioactive seeds, or reflector radars [12,13,14,15,16,17,18,19]. In addition to these labeling techniques, the tattooing of the SLN with carbon-based suspensions has recently been proposed. Tattoos are typically performed with commercial carbon-based colloids such as Spot^®^ colloids (GI Supply, Mechanicsburg, EE. UU.) and show high intraoperative identification rates (60-95%) and reduced false negative rates of 9.1–22% [20,21,22,23,24,25]. This technique is not free of limitations, as the reduced viscosity of those colloids restricts their preservation in the SLN, and these colloids often migrate into the surrounding adipose tissue, producing unwanted dissemination via the lymphatic route, which has been reported to occur in about 18% of cases [21,26]. In addition, it has been reported that different carbon-based colloids used for skin tattooing produced several inflammatory foreign-body-type reactions [27,28]. Although some authors suggest that carbon particles are inert and non-cytotoxic, other authors have observed inflammatory reactions of different degrees [29,30,31,32].

Novel approaches using nanostructures are available to identify SLN after NACT during surgery. For example, the magnetic detection of superparamagnetic tracers based on iron oxide (Fe_3_O_4_) nanoparticles has been widely reported [33,34,35,36]. The fluorescence emission of quantum dots has also been exploited for SLN localization, especially those based on cadmium selenide or telluride [37,38,39]. Taking advantage of fluorescence and photoacoustic properties, fluorescent dye-loaded mesoporous silica nanoparticles have also been used to mark lymph nodes [40]. Another approach has been the administration of Gd dendrimers to image the lymph node by magnetic resonance and optical imaging techniques [41,42]. However, these labels are performed for SLN detection after NACT and there is still a need to develop labels that persist in the lymph node for long periods of time and can be used for TAD.

Based on the above, in this study, we explored the use of melanin as a dye-labeling element encapsulated in polymer nanoparticles based on poly(lactic-co-glycolic acid) (PLGA) to render SLN tattooing suspensions to reduce the cytotoxicity associated with high doses of melanin [43]. Melanin is an endogenous pigment derived from the amino acid, tyrosine. Its most common formulation includes dihydroxyindole carboxylic acids and their reduced forms, commonly known as eumelanin [44]. In addition, not only to reduce its cytotoxicity at high doses but also to increase the viscosity of the labeling and increase the preservation of the marker in the lymph node, we encapsulated melanin in those polymeric PLGA nanoparticles for achieving a sustained release and an extended duration of its action. We performed histological evaluation and analyzed the efficacy of lymph node tattooing in a total of 7 pigs to study its biological behavior and tattooing efficiency over time (1, 2, 4, 6, and 16 weeks). 

## 2. Materials and Methods

### 2.1. Synthesis, Characterization and In Vitro Studies of Nanoparticles

#### 2.1.1. Materials

Poly(D,L-lactide-co-glycolide) (PLGA Resomer^®^ RG504H) was purchased from Evonik Industries AG (Essen, Germany), and sodium cholate, ethyl acetate and melanin (in powder) were purchased from Sigma Aldrich (Merck KGaA; Darmstadt, Germany). 

Human dermal fibroblasts (NHDF-Ad) were obtained from Lonza (Basel, Switzerland), J774A.1 mouse monocyte macrophages ATCC-TIB-67™ were acquired from LGC Standards (Barcelona, Spain) and the human breast adenocarcinoma cell line MDA-MB-231 was kindly donated by Dr. Alberto Jiménez Schumacher (Institute for Health Research Aragon, Zaragoza, Spain). These cell lines were used to determinate the subcytotoxic dose of the nanoparticles used.

#### 2.1.2. Synthesis of Nanoparticles with Encapsulated Melanin

Melanin encapsulation was carried out by the double emulsion solvent evaporation method (w/o/w) [45], as illustrated in Figure 1. In the first step, melanin was dissolved in the aqueous phase; for this purpose, a Milli-Q water solution was adjusted to pH 8 using NaOH, and melanin was added at varying concentrations (0.15, 0.3, 0.5 and 1 mg/mL). In parallel, different concentrations (5, 10, 15 and 20 mg/mL) of PLGA and 3 mL of ethyl acetate were used for the preparation of the organic phase. Then, 1 mL of NaOH–melanin was dissolved in the above solution, and 1 mL of Milli-Q water was added again. The mixture was sonicated at 40% amplitude for 20 s (Digital sonifier 450, Branson, MO, USA); 8 mL of 1% sodium cholate was added as surfactant and sonicated again at 40% for 40 s. Finally, 12 mL of 0.3% sodium cholate were added. The solution was left to evaporate for 3 h at 600 rpm on a shaking plate, and the samples were centrifuged at 7500 rpm and 15 °C for 15 min. Finally, the synthesized melanin-loaded nanoparticles (Mel-NPs) were resuspended at 10 or 20 mg/mL in a 1% (*w*/*v*) methylcellulose hydrogel for the preparation of suspensions.

#### 2.1.3. Electron Microscopy Studies

The morphological characterization and size distribution of the resulting particles were performed by scanning electron microscopy (SEM Inspect F50, FEI Company, Hillsboro, OR, USA) at an acceleration voltage of 5–10 kV. The particles were deposited on carbon tape placed on an aluminum slide. The samples were coated with a Pd layer using a high vacuum coater (Leica EM ACE200, Wetzlar, Germany). Diameter distributions were obtained from manual measurements using the free Image-J software (v1.52; National Institutes of Health, 2019) for a sample of 100 nanoparticles. The morphology and dimensions were also assessed using a T20-FEI transmission electron microscope (FEI Company, Hillsboro, OR, USA) at 200 kV. TEM samples were prepared by depositing 50 µL of the corresponding colloids dispersed in Milli-Q water on a formvar-coated copper grid and dried for at least 2 h.

#### 2.1.4. Quantification of Melanin Encapsulation

The quantification of the melanin encapsulated in the PLGA NPs was assessed by thermogravimetric analysis (Mettler Toledo TGA/STDA 851e, Mettler Toledo; Columbus, OH, US) using the degradation temperatures of the PLGA and melanin, analyzed from 30 to 800 °C with a N_2_ flow rate of 50 mL/min. This quantification was also indirectly performed by the evaluation of the UV–Vis spectra of the supernatants collected after the synthesis. The amount of melanin was quantified by UV–Vis spectroscopy (Jasco V670, Jasco, Easton, MD, USA), using the maximum absorbance at 193 nm attributed to the electronic transition of the melanin π-π* backbone. 

#### 2.1.5. In Vitro Release Studies

To obtain the melanin release profiles, Mel-NPs were dispersed in distilled water and kept for 28 days at 37 °C under continuous agitation. Aliquots of the supernatants were taken every day during the first week, and once a week for the rest of the month (1, 2, 3, 4, 5, 6, 7, 14, 21, and 28 days after the beginning of the experiment). The concentration of released melanin (in the supernatant) was obtained from the UV–Vis spectrum (absorbance maximum at 193 nm), as detailed in the previous section.

#### 2.1.6. Cell Viability Assays

The cytotoxicity of the Mel-NPs was determined using the Blue Cell Viability Assay (Abnova, Taipei, Taiwan) according to the manufacturer’s instructions. Briefly, the cells were seeded at concentration of 6000 cells/well for fibroblasts and J774 cells, and then 18,000 cells/well for the MBA-MD-231 cell line, before being incubated for 24 h at 37 °C and 5% CO_2_. Subsequently, Mel-NPs at different concentrations were added to the cells (0.1, 0.05, 0.025 and 0.01 mg/mL). After 24 h, the reagent (10% (*v*/*v*)) was added, cells were incubated for 4 h at 37 °C and 5% CO_2_ and fluorescence were read at 530/590 ex/em in a Varioskan LUX microplate reader (Thermo Fisher Scientific, Waltham, MA, USA). Cytotoxicity was evaluated by comparing the values obtained for the treated cells with those retrieved from untreated cells, assigning to these controls 100% viability. Four replicas of each experiment were performed in triplicate. All results are presented as mean ± standard deviation. Data were analyzed using two-way analysis of variance (ANOVA) (GraphPad Prism 8, San Diego, CA, USA). Statistically significant differences were considered when *p* ≤ 0.05.

### 2.2. In Vivo Studies of Mel-NPs

#### 2.2.1. Animals and Surgical Procedures

Seven female White Landrace pigs with an average weight of 25 kg (ranging from 22 to 30 kg) were used. The experimental animals were divided into two study groups. The “short-term study” included 4 animals that were housed for 1, 2, 4 and 6 weeks, and the “long-term study” included 3 animals that were housed for 16 weeks. The study was carried out in the facilities of the Institute for Health Research Aragon (Zaragoza, Spain), in accordance with the Spanish Policy for Animal Protection RD53/2013 which meets the European Union Directive 2010/63. The experimental procedure was approved by the Animal Research Ethics Committee of the University of Zaragoza under the reference PI09/20.

In all animals, injections of the pigments were administered in selected peritoneal lymph nodes from the splenic artery root, portal vein root and distal ileum regions. A total of 10 lymph nodes were tattooed with Mel-NPs (20 mg/mL) in the “short-term studies”. In the “long-term studies”, 6 lymph nodes were tattooed with 20 mg/mL of Mel-NPs, and 9 lymph nodes were marked with 10 mg/mL of Mel-NPs to also analyze the concentration influence. The surgical procedure was performed under general anesthesia with oral intubation, mechanical ventilation and neuromuscular blockade. Animals were premedicated with zolazepam (Zoletil, Virbac, Hong Kong, 0.05 mg/kg) and intramuscular dexmedetomidine (Dexmopet, Fatro Iberica SL, Sant Just Desvern, Spain, 0.08 mL/kg). Anaesthetic induction was performed with disopropylphenol (Propofol 1% MCT, Fresenius Kabi Laboratories Spain, Barcelona, Spain, 6 mg/kg) and sevofluorane 1.9% (Baxter SL, Valencia, Spain) was used for maintenance. Muscle block was induced with pancuronium bromide (Pavulon, Organon Española SA, Jersey City, NJ, USA, 4 mg/mL). Intraoperative analgesia consisted of the continuous infusion of fentanyl (Fentanest, Aurovitas España, Madrid, Spain, 10 μg/kg/h), and Ringer’s lactate (8 mL/kg/h) was the solution used as fluid therapy. For vector administration surgery, a midline laparotomy was performed to expose the abdominal viscera and isolate the selected mesenteric nodes. Vectors were directly administrated in the lymph nodes (2 or 3 lymph nodes per pig) in 0.2 mL doses (10 and 20 mg/mL) loaded into 1 mL syringes through 21G needles. After surgery, the animals were housed under special conditions in heated cages. We used buprenorphine (Buprenodale, Dechra, Northwich, UK, 0.05–0.1 mg/kg/day) during the first 72 h after surgery, and antibiotic prophylaxis with enrofloxacin (Enroflox, Agrovet Market SA, Lima, Peru, 2.5 mg/kg) until the lymph node dissection surgery was performed after 1, 2, 4, 6 or 16 weeks. For lymph node dissection surgeries, a wide median laparotomy was also performed, and the tattooed lymph nodes were resected. At the end of the experiment, the animals were euthanized by a single injection of potassium chloride (1 mEq/kg) intravenously, preventing depolarization of the heart muscle and causing it to stop.

#### 2.2.2. Histopathological Studies

After sampling, lymph nodes were fixed in 4% formalin for 3–5 days and cut into approximately 3 mm thick tissue slices. The most pigmented slice was selected for histological processing. Samples were processed according to standard procedures and stained with hematoxylin and eosin (H&E) for light microscopy examination by Scientific Technical Services—Microscopy and Pathology from the Institute for Health Research Aragon.

#### 2.2.3. Statistical Analysis

This study included 1 macroscopic variable (the ‘ease of identification’) and 3 microscopic variables (the ‘location of the nanoparticles’ and the ‘extent of inflammation’). Study groups were made according to time (‘short term’ and ‘long term’) and Mel-NPs concentration (10 and 20 mg/mL). To study the difference between groups, the Mann-Whitman U test was performed using IBM SPSS Statistics 29.0.10 (SPSS Inc.; Chicago, IL, USA). Statistically significant differences were considered when *p* ≤ 0.05.

Ease of identification. Before dissection surgery, the tattooed lymph nodes were evaluated in situ and the “ease of visual identification” was subjectively assessed by experienced surgeons on a scale from 1 to 3 (1 being ‘indistinguishable’ and 3 “optimal”), as depicted in Figure 2a.Location of the nanoparticles. The determination of the location of the particles was made subjectively according to the predominant location of the marking (Figure 2b). As shown in Figure 2b, the location of the marking was classified as: inner zone, pericapsular region, adjacent tissue, inner zone/pericapsular region, inner zone/adjacent tissue, pericapsular region/adjacent tissue or all locations.Extent of inflammation. The degree of intensity of the inflammatory response was quantified according to the extent of the inflamed tissue. The most extensive section of inflammatory tissue in the lymph node was manually measured using the ImageJ software 2024 [46].

#### 2.2.4. Ultrastructural Studies

After sampling, biopsies were fixed in 2% glutaraldehyde in PB for 3 days and subsequently washed with PB. Samples were post-fixed with 2% osmium, rinsed, dehydrated in graded acetone (30%, 50%, 70% with 2% uranyl acetate, 90%, 100%), cleared in propylene oxide and embedded in araldite (Durcupan, Fluka AG; Buchs SG, Switzerland, Hatfield, PA, USA). A RMC MT-XL ultramicrotome was used for obtaining semi-fine and ultra-fine tissue slices. Semi-thin (1.5 μm) and ultra-thin (0.05 μm) sections were cut with a diamond knife. Semi-thin sections were stained with 1% toluidine blue and examined by light microscopy (Olympus BX51 microscope, Olympus Imaging Corporation; Tokyo, Japan). Ultra-thin sections were collected on Formvar-coated single-slot grids counterstained with 1% uranyl acetate and Reynold’s lead citrate staining. The samples were observed at the Electronic Microscopy Service of Biological Systems of the University of Zaragoza with a JEOL JEM 1010 transmission microscope (JEOL Ltd., Tokyo, Japan) operating at 80 kV. 

## 3. Results

### 3.1. Synthesis and Characterization of Melanin-Loaded Nanoparticles

#### 3.1.1. Optimization of PLGA and Melanin Concentration

To analyze the effect of PLGA concentration on particle size, several syntheses were performed by varying the amount of polymer (without encapsulating melanin). As illustrated in Figure 3a, particle size increases with PLGA concentration, attributed to an increased viscosity of the organic solution. This observation is in agreement with previous results, and it is corroborated by the increase in the viscosity of the organic phase along with the increase in PLGA concentration [47]. All colloids were homogeneous in their particle size distribution (with a polydispersity index, PDI < 0.01) [48]. It was decided to continue the optimization process using PLGA at a concentration of 10 mg/mL, since this minimizes the use of polymer and no significant difference in the resulting NPs sizes was found. On the other hand, an attempt was made to obtain colloids with high melanin load, as melanin provides the macroscopic coloring necessary for the visualization of lymph nodes with the naked eye. Four different syntheses were performed under varying melanin concentrations (0.15, 0.3, 0.5 and 1 mg/mL). The maximum amount of melanin at 1 mg/mL was set as the solubility limit for achieving its total dissolution in Milli-Q water (pH 8). Increasing the pH of the water to pH 8 provides an environment in which melanin can be properly ionized, increasing its solubility and allowing a better dispersion in water. In all syntheses, the colloids were homogeneous in their particle size distribution (PDI < 0.01). As depicted in Figure 3b, given the morphological similarity of the resulting Mel-NPs, the emulsion with the highest melanin concentration (1 mg/mL) was chosen to subsequently obtain a superior macroscopic staining. 

#### 3.1.2. Characterization of the Optimized Mel-NPs

The morphology of the optimized Mel-NPs was analyzed using TEM. As shown in Figure 4a, melanin can be observed as a dark pigmentation in the PLGA matrix, indicating a successful melanin encapsulation. The optimized Mel-NPs measured 117 ± 38 nm on average, with a relative standard deviation (RSD) of 32.7% and a PDI < 0.01. 

An attempt was made to quantify the encapsulation efficiency by TGA, but Mel-NPs were difficult to differentiate from PLGA since melanin starts to degrade at low temperatures and overlaps with the degradation curve of polymers. However, since the TG curve of Mel-NPs does not completely pyrolyze at 450 °C, it was possible to establish a minimum melanin concentration of 17.8% (Figure 4b). To corroborate these findings, the concentration of encapsulated melanin was quantified indirectly through the UV–Vis spectrum of the supernatant knowing the melanin absorbance maximum at 193 nm (Figure 4c). With this method, a melanin encapsulation yield of 65.8 ± 9.1% (with respect to the initial concentration used in the synthesis), which represents 23.5% melanin loading by weight in the resulting Mel-NPs.

In vitro melanin release was determined by the UV–Vis spectrophotometry of the supernatant. For the experiment, Mel-NPs were dispersed in distilled water and kept for 28 days at 37 °C under continuous agitation. As shown in Figure 4d, Mel-NPs exhibits an initial burst release due to the unencapsulated weakly adhering melanin on the nanoparticle surface, and after 4 weeks, ∼24% melanin was continuously released from the NPs synthesized with PLGA. PLGA is a biodegradable polymer that is degraded due to the hydrolysis of its ester linkages, leading to the gradual release of the drug by diffusion as the polymer matrix erodes.

The cytotoxicity of Mel-NPs was determined in three cell lines: MDA-MB-231 (human breast adenocarcinoma), J774 (murine macrophages) and NHDF-Ad (human dermal fibroblasts), 24 h after incubation. As shown in Figure 4e, Mel-NPs were not cytotoxic at any concentration tested in any of the studied cell lines (79–100%), which are closely related to the intended clinical application. The results are in compliance with ISO 10993-5, which states that cell viability higher than 70% is considered as cytocompatibility [49].

### 3.2. Intraoperative Identification of Tattooed Lymph Nodes

As shown in Figure 5, a total of 25 nodes were tattooed with Mel-NPs-based vectors (10 lymph nodes using 20 mg/mL in the short-term studies; 6 lymph nodes using 20 mg/mL and 9 lymph nodes using 10 mg/mL in the long-term studies). The short-term studies consisted of stabling for 1-6 weeks, while the long-term studies were conducted 16 weeks after the administration. We did not observe any migration of the tattoo to other lymph nodes in any case, so the correlation between the tattooed lymph node and the lymph node established as ‘sentinel’ corresponds to 100%. As depicted in Figure 5b, 77.8–83.3% of the tattooed nodes were identified during the surgery (identification rate). Vector identification rates of Mel-NPs (20 mg/mL) were similar in the short- and long-term studies, suggesting that tattooing is stable for up to 16 weeks. Additionally, in both short-term and long-term studies, the nodes clearly showed visible tattoos (Figure 5c). Differences in concentration of Mel-NPs were observed, reducing the identification rate by about 30% (Figure 5d). The ease of identification was also notably reduced (Figure 5e and Appendix A). However, the difference was not statistically significant.

### 3.3. Histological Studies

#### 3.3.1. Histopathological Findings

In all tattooed lymph nodes, granulomatous foreign-body-reaction-type inflammation associated with the presence of Mel-NPs was identified. The inflammatory process was characterized by the presence of macrophages and multinucleated giant cells that phagocytosed and isolated the Mel-NPs from the first week after inoculation (Figure 6a). Necrotic foci associated with the foreign-body-type inflammatory reaction to the administration of the particles were observed in some lymph nodes (Figure 6b).

#### 3.3.2. Location of the Nanoparticles

A histopathological evaluation of the labelled lymph nodes revealed that the particles were mainly located in the internal region of the node, in the pericapsular area or in the tissue adjacent to the node, although occasionally mixed particles were found (Figure 7). In the short-term studies, Mel-NPs (20 mg/mL) were mainly found in the pericapsular region (50%) and in the adjacent tissue (35.7%). In 14.3% of cases, they were also found in the interior of the lymph node. In the ‘long-term’ studies, Mel-NPs were not found inside the lymph node and were present in the adjacent tissue (44.4%) and in the pericapsular region (55.6%). On the other hand, Mel-NPs at a 10 mg/mL concentration (in the long-term studies) were found in the pericapsular region (36.4%), in the adjacent tissue (36.4%) and inside the lymph node (27.3%) (Appendix A). No statistically significant differences were detected. 

#### 3.3.3. Extent of Inflammation 

The extent of inflammation produced by Mel-NPs (20 mg/mL) in the ‘short-term studies’ was 398 ± 290 µm, while in the ‘long-term studies’ it was 520 ± 223 µm (Appendix A). Again, there was a considerable variability between samples. In contrast, in the long-term studies, a substantial reduction in inflammation was observed in Mel-NPs-labelled lymph nodes at 10 mg/mL (278 ± 139 µm), although this was not statistically significant.

### 3.4. Ultrastructural Study

In the ultrastructural study, Mel-NPs were mainly found internalized in macrophages. Macrophages initiate phagocytosis by forming filopodia and lamellipodia to capture particles from the extracellular space (Figure 8a). Once captured, macrophages internalize the particles into their cytoplasm within phagosomes. Inside the phagosomes or phagolysosomes, Mel-NPs can be identified. In the ‘short-term studies’, the average size of the Mel-NPs was 264 ± 97 nm (Figure 8b,c), which reduced to 43 ± 13 nm in ‘long-term studies’ (Figure 8d and Appendix A). This represents a reduction of approximately ∼3.4 times their size. However, in all cases, particle aggregation is observed, resulting in large clusters of Mel-NPs. Especially in the long-term studies, numerous lysed macrophages were observed, with broken plasma membranes and free organelles released into the extracellular space. This suggests that residues of crystallized melanin, with sharp edges, could break the membrane of the macrophages’ phagolysosomes and lead to cell lysis (Figure 8e).

## 4. Discussion

The marking of the SLN through tattooing with carbon-based suspensions is increasingly used in breast cancer patients prior to NACT. To date, more than 800 SLN-labeling studies have been reported in various clinical trials [20,21,22,23,24,25,50,51,52,53,54,55], suggesting that, despite its apparent advantages over other methods, its use has not yet become widespread in the majority of healthcare centers. This strategy is low-cost, increases identification rates compared to other techniques (i.e., surgical clips, magnetic seeds, radioactive seeds, reflector radars, etc.), does not require specialized equipment, and minimizes additional discomfort for the patient, making it a promising alternative for SLN marking in the field of oncological surgery [56]. However, the materials available for SLN tattooing are mainly limited to carbon-based suspensions (i.e., Spot^®^ (Mechanicsburg, PA, USA), CARBO-REP^®^ (Sterylab, Milan, Italy) or Black Eye^®^ (The Standard Co., Ltd., Gyeonggido, Republic of Korea)) used for other clinical purposes (such as external marking tumors for surgical resection). Therefore, the development of specific materials showing enhanced marking efficacy is required for the optimization of the technique.

For this purpose, we propose the use of melanin as a tattooing element to replace carbon-based colloids, since as an endogenous pigment, it could potentially reduce associated cytotoxicity [43]. Additionally, melanin exhibits therapeutic properties including antioxidant, photoprotective, anti-inflammatory, and anti-tumoral in melanomas [57]. We estimate that approximately 23.5% by weight of the optimized Mel-NPs is melanin, and that about 24% of melanin is released in vitro after 4 weeks. In other studies on the release of melanin from PLGA NPs (LA:GA 50:50), releases of 85% have been observed after 25 days under similar test conditions [58]. The rate of polymer degradation is subjected to the physicochemical characteristics of the NPs and the conditions of the assay [59]. However, the Mel-NPs synthesized in this study had lower release rates of melanin compared to previous reports. According to previous studies [57], melanin-loaded NPs did not show cytotoxicity in MDA-MB-231 (breast adenocarcinoma) cell lines, fibroblasts, and J774 (macrophages), with viabilities between 79 and 100%. In our case, we did not observe any cytotoxicity at the doses tested. 

In the in vivo studies, after the stabilization period for each group of animals, we simulated the TAD technique for the selective excision of each lymph node. In short-term studies (1–6 weeks), the intraoperative identification rate was 77.8%, while in long-term studies (16 weeks), the identification rate was 83.3%. These results are similar to those reported in other clinical and preclinical studies conducted with carbon-based suspensions, where average identification rates of 91.2 ± 11.7% (64–100%) have been described [20,21,22,23,24,25,50,51,52,53,55]. However, in our study, we used 0.2 mL of Mel-NPs, whereas other studies use variable doses ranging from 0.1 to 1 mL. Additionally, in previous clinical trials a correlation between the tattooed lymph node (TLN) and the SLN of 81.9 ± 18.6% (47.9–100%) was described [20,21,22,23,24,25,50,51,52,53,55]. In our study, we found no dissemination to other lymph nodes in any case, so 100% of the SLN would have corresponded to the TLN.

In all the dissected lymph nodes, a foreign-body-type inflammatory reaction was found of varied degrees, characterized by the presence of macrophages and multinucleated giant cells that phagocytosed and isolated the particles inoculated in the node. This type of inflammation associated with the inoculated particles has been previously described in skin tattoo [27,28]. Additionally, in some cases, we found small necrotic foci associated with inflammation. However, in skin tattoos, extensive necrosis has only been described, primarily associated with infections [60,61]. Interestingly, the inflammatory reaction did not decrease over time; instead, it seemed to be more intense after 16 weeks. In our ultrastructural studies, we observed that the melanin loaded on PLGA NPs disrupts macrophage phagosomes, leading to cell lysis. In long-term studies, more non-encapsulated melanin residues were found, suggesting that the polymer degrades and releases the melanin, which appears to be more toxic to cells. Indeed, we observed that Mel-NPs reduced in size by ∼3.4 times relative to long-term studies. Although in vitro release was less than 25% after 4 weeks, the degradation rate of the polymer is dependent on the pH of the medium, so that the PLGA degradation of the NPs/MPs is accelerated by the presence of the acidity of the lysosomal enzymes and in vivo conditions. Indeed, under recreated lysosomal conditions, PLGA degradation is estimated to be close to ∼40% after 4 weeks, which is in agreement with our in vivo observations [62,63].

## 5. Conclusions

Mel-NPs showed high identification rates and high scores on ease of identification both at 10 and 20 mg/mL, suggesting that 10 mg/mL is sufficient to mark the lymph nodes. Although not statistically significant, it would be expected that more NPs in the lymph node would produce greater inflammatory reactions. Reducing the dose of foreign material in the body is therefore of interest. In our long-term studies, we observed that much of the encapsulated melanin was released, leading to an increased presence of melanin residues. These residues, under the electron microscope, appeared rigidly shaped and edged, making them look crystallized. We observed that these geometries led to cell death by disruption of the phagolysosomes. Therefore, using smaller materials and a slower sustained release of polymers may be a promising approach for the application of sentinel lymph node tattooing.

## Figures and Tables

**Figure 1 nanomaterials-14-01149-f001:**
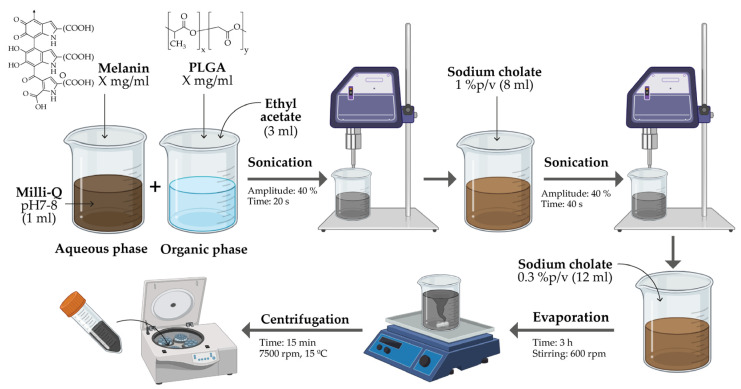
Schematic representation of the synthesis of melanin-loaded PLGA nanoparticles by the water-in-oil-in-water (w/o/w) method.

**Figure 2 nanomaterials-14-01149-f002:**
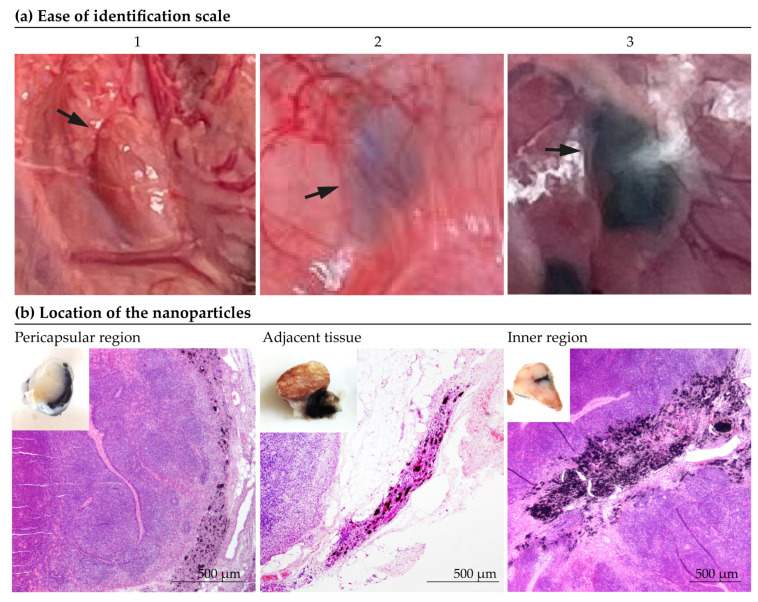
Determination of the ease of identification and the location of the particles. (**a**) Ease of identification scale. (**b**) Examples of location of the nanoparticles.

**Figure 3 nanomaterials-14-01149-f003:**
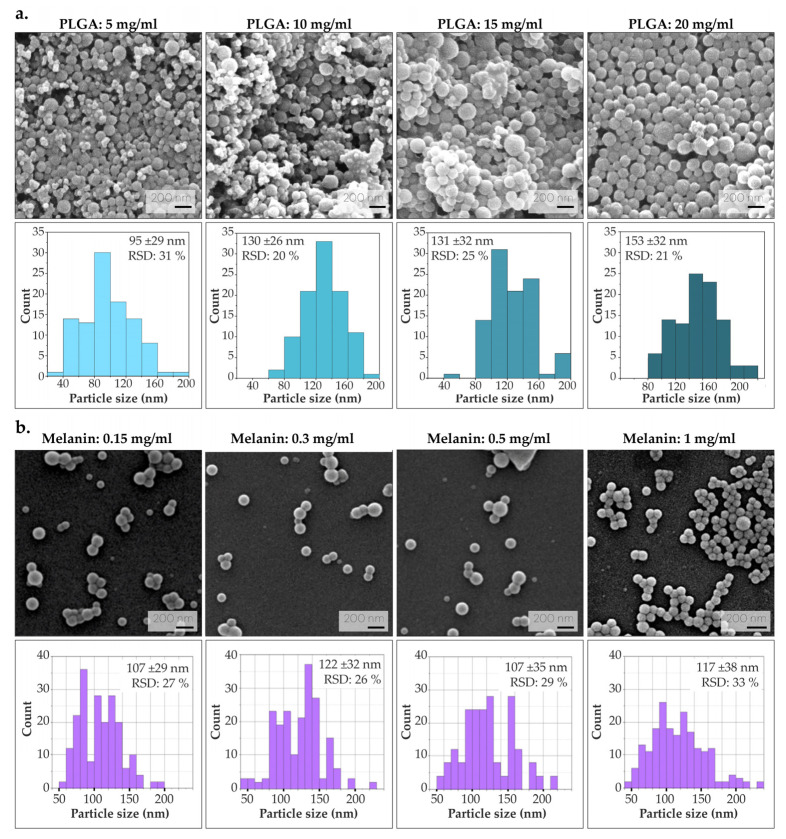
Morphology of the NPs during synthesis optimization, SEM micrographs are depicted together with their respective diameter distribution histograms. (**a**) Optimization of PLGA concentration (5, 10, 15 and 20 mg/mL). (**b**) Optimization of melanin concentration (0.15, 0.3, 0.5 and 1 mg/mL) using PLGA at 10 mg/mL.

**Figure 4 nanomaterials-14-01149-f004:**
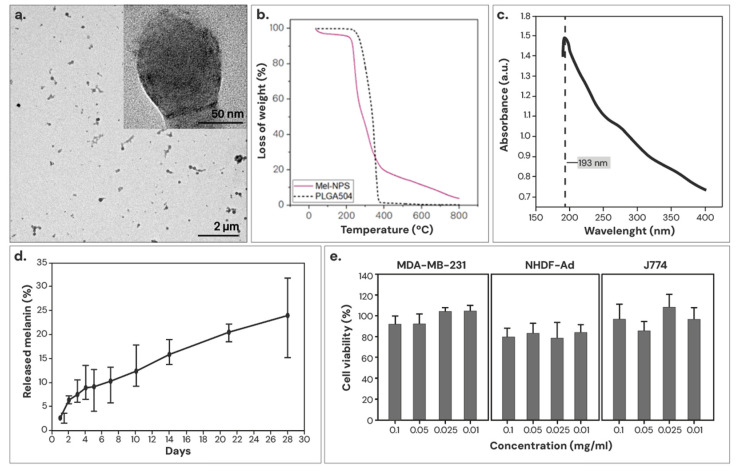
Characterization of the Mel-NPs. (**a**) Transmission electron micrographs of the particles, where melanin is clearly distinguished by its higher electron density. (**b**) Thermogravimetric curves. (**c**) UV–Vis absorbance spectrum of melanin, with a maximum of absorbance at 193 nm. (**d**) Curve of the release kinetics of the encapsulated melanin from the Mel-NPs in distilled water. (**e**) Cell viability in the human breast adenocarcinoma (MDA-MB-231), macrophages (J774), and fibroblasts (NHDF-Ad) cell lines. Cytotoxicity is determined by assigning 100% cell viability to untreated control cells.

**Figure 5 nanomaterials-14-01149-f005:**
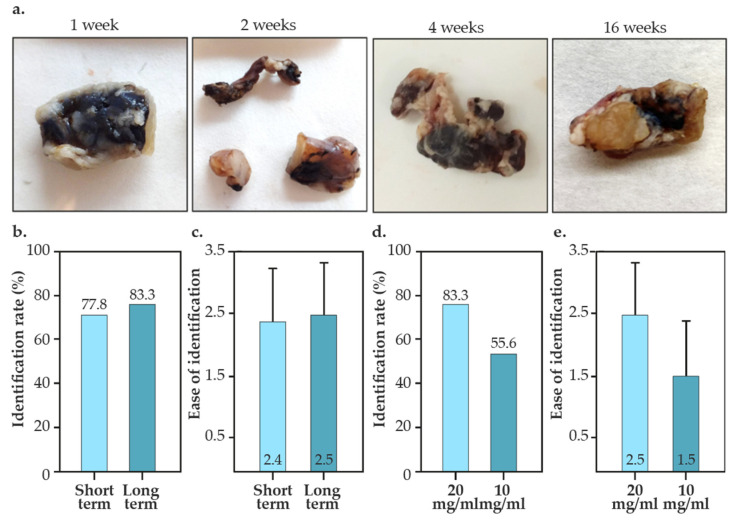
Identification rate and ease of identification of Mel-NPs tattooed lymph nodes. (**a**) Photographs of lymph nodes tattooed with Mel-NPs vectors (20 mg/mL) at different times. (**b**) Identification rates of lymph nodes tattooed with Mel-NPs vectors (20 mg/mL) in the short- and long-term studies. (**c**) Ease of identification of lymph nodes tattooed with Mel-NPs vectors (20 mg/mL) in the short and long-term studies. (**d**) Identification rates of lymph nodes tattooed with Mel-NPs vectors using 20 and 10 mg/mL colloids in the long-term studies. (**e**) Ease of identification of lymph nodes tattooed with Mel-NPs vectors using 20 and 10 mg/mL colloids in the long-term studies.

**Figure 6 nanomaterials-14-01149-f006:**
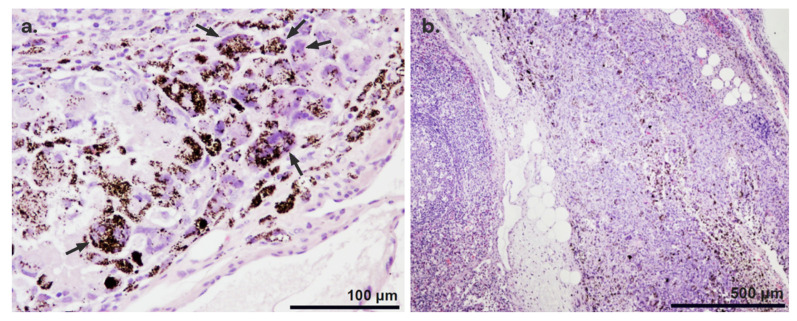
Histopathological findings of tattooed lymph node of Mel-NPs, where Mel-NPs appear brown and are easily distinguished from H&E staining. (**a**) The distribution of Mel-NPs shows a quite homogeneous morphology characterized by small granules. A foreign body reaction associated with Mel-NPs was observed, characterized by the presence of multinucleated giant cells (arrows). NP-loaded macrophages are also depicted. (**b**) Foci of necrosis associated with a foreign body reaction were observed in the ‘short-term studies’.

**Figure 7 nanomaterials-14-01149-f007:**
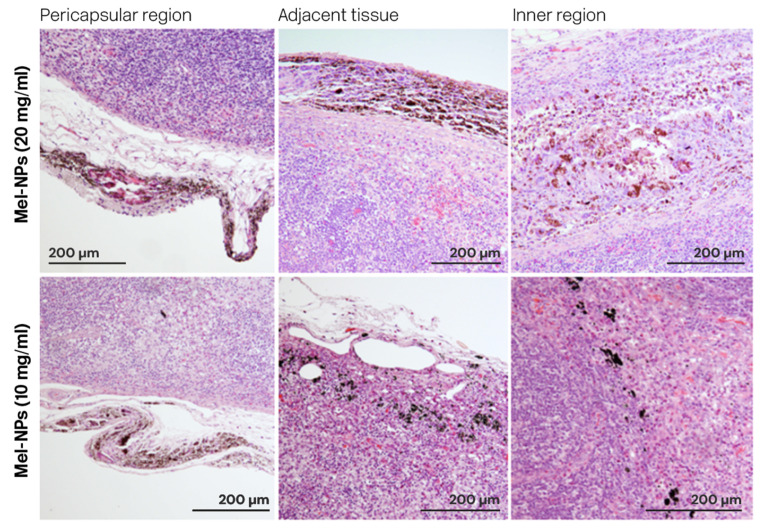
Location of Mel-NPs (10 and 20 mg/mL) in the adjacent tissue, the pericapsular region, and the inner lymph node region.

**Figure 8 nanomaterials-14-01149-f008:**
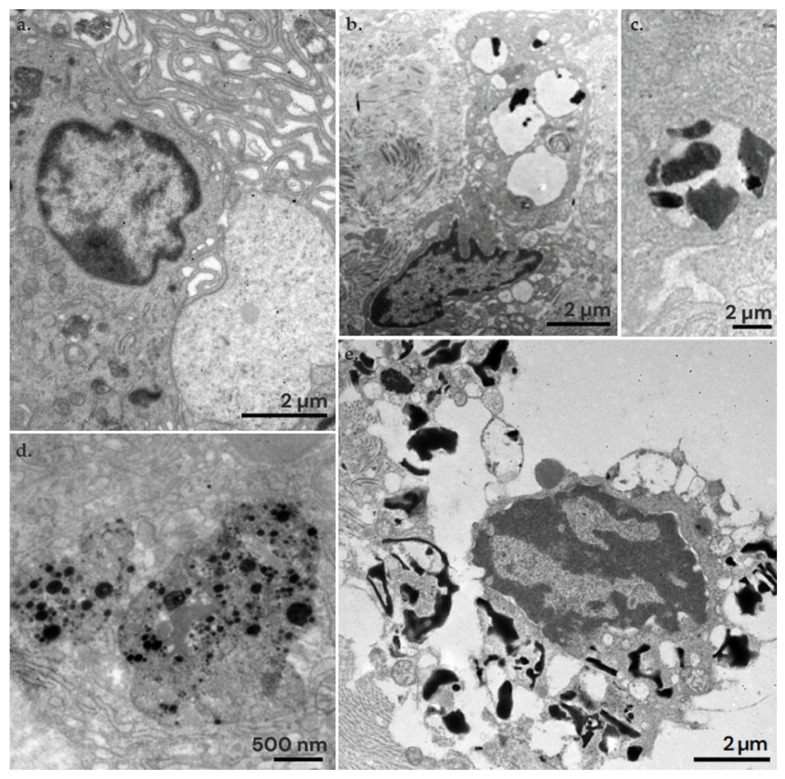
Ultrastructural findings in the lymph nodes tattooed with Mel-NPs at 20 mg/mL. (**a**) Activated macrophages with numerous lamellipodia and filopodia for the phagocytosis of Mel-NPs. (**b**) Macrophage with Mel-NPs in its cytoplasm. (**c**) Detail of Mel-NPs within the phagosome of the macrophage in the ‘short-term studies’. (**d**) Detail of Mel-NPs within the phagosome in the ‘long-term studies’, where the particles are more electron-dense and smaller in size. (**e**) A macrophage lysed by the action of the Mel-NPs. As observed, large amounts of non-encapsulated and potentially crystallized melanin appear that break the membranes and lead to cell death.

## Data Availability

Data are contained within the article and Appendix A.

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
