# Peer review of "Melanin-Based Nanoparticles for Lymph Node Tattooing: Experimental, Histopathological and Ultrastructural Study"

_nanomaterials, 2024, doi:10.3390/nano14131149_

Round 1
Reviewer 1 Report
Comments and Suggestions for Authors
The manuscript titled "Melanin-based Nanoparticles for Lymph Node Tattooing: Experimental, Histopathological and Ultrastructural Study" explores the use of melanin-loaded polymeric nanoparticles (Mel-NPs) for lymph node tattooing in breast cancer. The study aims to overcome limitations associated with carbon-based colloids, such as inflammatory reactions and migration issues. The design of this study is new, and all the experiments, from nanomaterials preparation, characterization, biological evaluation, and histological analysis, are well-organized and presented. Overall, this manuscript has sufficient significance and quality for publishing. However, some issues should be resolved to improve it further.
1. Please change all the spelling of "tumour" to "tumor", "labelling" to "labeling", in this manuscript, as "tumor" and "labeling" is the preferred spelling in American english and are more frequently seen in the literature.
2. The references in this manuscript are somehow out-of-date. Please try to update some of them with new literature, especially the references 2-8 in the introduction part.
3. The introduction part does not include the recent development of nanomaterials in SLN tattooing. Please briefly discuss.
4. I like the illustration of the synthesis of melanin-loaded PLGA nanoparticles in Figure 1. However, it would be better to have the chemical structures of both PLGA and melanin in this schematic representation.
5. Figure 3, please adjust the scale bar of the SEM imaging to 100 or 200 nm levels, which is the average size of the nanoparticles. The current scale bar is oversized.
6. In Figure 4C and Line 146, why measure the absorption wavelength of melanin at 193 nm? This is a short UV wave and usually not suitable for UV quantification because of the high background here. Melanin absorbs broadly from the UV to near-infrared spectrum; thus any wavelength >400 nm may be better because of the low background. Ref: Nat Commun 11, 4569 (2020).
7. In Figure 4a, why is the TEM imaging not very clear, and why do the nanoparticles look so irregular?
8. In Figure 4e, a positive control should be included for the cell viability assay.
9. Figure 5: why does this experiment have no negative control? In addition, the numbers for Figure 5c~e in the caption should be corrected.
10. In Figure 6, the numbers of a, b are missing. The scale bar for Figure 7 is missing.
11. For histopathological evaluation of the labeled lymph nodes in Figure 7, the control group treated with non-melanin-loaded PLGA should be included. The quantification method of the labeled lymph nodes for this figure is not clearly stated.
Author Response
The manuscript titled "Melanin-based Nanoparticles for Lymph Node Tattooing: Experimental, Histopathological and Ultrastructural Study" explores the use of melanin-loaded polymeric nanoparticles (Mel-NPs) for lymph node tattooing in breast cancer. The study aims to overcome limitations associated with carbon-based colloids, such as inflammatory reactions and migration issues. The design of this study is new, and all the experiments, from nanomaterials preparation, characterization, biological evaluation, and histological analysis, are well-organized and presented. Overall, this manuscript has sufficient significance and quality for publishing. However, some issues should be resolved to improve it further.
We sincerely appreciate his/her positive assessment of our manuscript. His/her suggestions definitively and substantially strengthen the overall quality of the manuscript. We thank the reviewer for recommending our work for publication.
Comment 1: Please change all the spelling of "tumour" to "tumor", "labelling" to "labeling", in this manuscript, as "tumor" and "labeling" is the preferred spelling in American english and are more frequently seen in the literature.
Response 1: We thank the reviewer for his/her suggestions. These words have now been updated in the revised version of the manuscript.
Comment 2: The references in this manuscript are somehow out-of-date. Please try to update some of them with new literature, especially the references 2-8 in the introduction part.
Response 2: We have now added new updated references following the reviewer's recommendations, for example see: (5), (6), (11), (17), (18) and (19).
Comment 3: The introduction part does not include the recent development of nanomaterials in SLN tattooing. Please briefly discuss.
Response 3: We agree. We have now added a brief discussion on the development of new nanomaterials for ganglion tattooing on lines 78-87. However, these techniques are only used for labelling the SLN prior to surgery, and, to the best of our knowledge, there are no new approaches for TAD. This reinforces the interest of this work.
Comment 4: I like the illustration of the synthesis of melanin-loaded PLGA nanoparticles in Figure 1. However, it would be better to have the chemical structures of both PLGA and melanin in this schematic representation.
Response 4: We agree with this suggestion. We have redesigned the image by updating the chemical structures of the PLGA and melanin used.
Comment 5: Figure 3, please adjust the scale bar of the SEM imaging to 100 or 200 nm levels, which is the average size of the nanoparticles. The current scale bar is oversized.
Response 5: We have adjusted the scale bars to 200 nm. We think it now displays better. Thanks for the suggestion.
Comment 6: In Figure 4C and Line 146, why measure the absorption wavelength of melanin at 193 nm? This is a short UV wave and usually not suitable for UV quantification because of the high background here. Melanin absorbs broadly from the UV to near-infrared spectrum; thus any wavelength >400 nm may be better because of the low background. Ref: Nat Commun 11, 4569 (2020).
Response 6: We thank the reviewer for his/her comment. Absorbance at 193 nm was measured for the quantification of melanin because it exhibits a clear absorption peak, as indicated in the figure below.

Also we used increased melanin concentrations to obtain the calibration curve:

We subtracted baseline to avoid any interference and by measuring the supernatant the melanin released was quantified due to the lack of interferences with the PLGA-based encapsulating matrix.
Comment 7: In Figure 4a, why is the TEM imaging not very clear, and why do the nanoparticles look so irregular?
Response 7: The reviewers have highlighted this concern, and we agree with his/her concern. The TEM image is not of sufficient quality to be published in the form presented. The reason for the irregularity is that the sample was negatively stained for contrast enhancement, and the contrast agent itself generated artifacts. A new TEM image is proposed in the revised version of the manuscript where it can be observed a low magnification image to convince the reviewers that the nanoparticles are homogenous and an inset including a high magnification image where a PLGA nanoparticle and inner melanin nanodeposits (as dark shadows inside) are depicted.

Comment 8: In Figure 4e, a positive control should be included for the cell viability assay.
Response 8: As explained in the materials and methods section (lines 166-168), cytotoxicity was assessed by comparing the values obtained for treated cells with those recovered from untreated cells, assigning 100% viability to these controls. So, 100% viability was obtained and assigned from the control cells. To clarify this protocol for the reader, we have added " Cytotoxicity is determined by assigning 100% cell viability to untreated control cells " to line 312 of the results section. We have not incorporated PLGA nanoparticle controls (without encapsulated melanin) because the cytotoxicity of PLGA nanoparticles has been extensively studied on various cell lines in previous work, consistently demonstrating that it is a biocompatible, non-cytotoxic polymer (DOI: 10.2217/nnm-2018-0258, 10.1016/j.ijbiomac.2024.132563).
Comment 9: Figure 5: why does this experiment have no negative control? In addition, the numbers for Figure 5c~e in the caption should be corrected.
Response 9: The lymph node identification experiment has no negative control because the non-tattooed nodes are the negative control. That is, the PLGA without encapsulated melanin would not generate contrast and therefore cannot be used as a negative control. Since the aim of the project is to identify the tattooed lymph node, we have used the other nearby lymph nodes that have not been tattooed as the negative control. Thank you, we have corrected letters (d) and (e) in the figure. We apologize for the error.
Comment 10: In Figure 6, the numbers of a, b are missing. The scale bar for Figure 7 is missing.
Response 10: Thank you for pointing out these errors. They have been now corrected.
Comment 11: For histopathological evaluation of the labeled lymph nodes in Figure 7, the control group treated with non-melanin-loaded PLGA should be included. The quantification method of the labeled lymph nodes for this figure is not clearly stated.
Response 11: We have not included a control group of just unloaded PLGA in the pig lymph nodes because they would not generate any contrast and therefore the nodes could not be identified. However, we did not discuss melanin-associated inflammation in relation to that of PLGA, since melanin alone probably could not be injected because of its rapid diffusion (and thus lymphatic migration) rates, which is one of the current problems with free carbon colloids. In Figure 7 we only show where the particles are located in the lymph node, as explained in Figure 2 in the methods section.
Thank you again.

Reviewer 2 Report
Comments and Suggestions for Authors
The submitted manuscript describes the development of melanin pigment-containing nanoparticles for the detection of lymph nodes. Although the research object of the submitted manuscript is interesting and the contents and quantity of experiments seem to be sufficient, the authors do not show a sufficient and persuasible explanation for each experiment. The authors should rebuild the last paragraph of Introduction part because it does not reflect the contents of the experimental results. And, the authors should clearly state their opinion for reference [29]. Based on these points, the reviewer judges that the submitted manuscript should be revised largely for publication for “nanomaterials”.
What is PDI? (line 250) The authors should show the basis of “PDI < 0.01”. The reviewer cannot understand the meaning of the sentence ”the particle size did not vary substantially from one synthesis to another, although it did increase as the PLGA concentration increased.” (line 246 and 247). Also, the reviewer cannot understand the meaning of the sentence “On the other hands, an attempt was made to obtain colloids with high melanin-load Mel-NPs.”. (line 253) The authors should describe the intention of showing the pH of Milli-Q water. The reader cannot see the scale bar in Figure 3a. The authors should explain why the particle size of Mel-NPs decreases in the presence of melanin. While Figure 3a and Figure 3b show clear SEM images of synthesized particles, Figure 4a shows the indecisive TEM image of them. If the synthesized Mel-NPs are not so fragile, the authors should show the robustness of them. It is difficult to evaluate the amounts of melanin encapsulated in Mel-NPs by the TG curve shown in Figure 4b. Since Figure 4c just indicates the UV spectrum of Mel-NPs, Figure 4c does not indicate the amount of melanin encapsulated in Mel-NPs. The authors must show the UV spectrum of PGLA particles. Based on these points, the reviewer thinks that the amounts of melanin encapsulated in Mel-NPs cannot be estimated from Figure 4b and Figure 4c. Additionally, the reviewer cannot understand the meaning of the sentence “a melanin encapsulation yield of 65.8 ± 9.1 %, which represents 23.5 % loading by weight in the particles.”. (line 278 and 279) The sentence “For the experiment, Mel-NPs were dispersed in PBS 1x and kept for 28 days at 37°C under continuous agitation.” (line 281 and 282) is not clear. And, the sentences from line 282 to 287 suggest the synthesized Mel-NPs are fragile in PBS. If so, the reviewer thinks that the synthesized Mel-NPs will not be available for any bioassay. The description on the cytotoxicity of the synthesized Mel-NPs using cell lines is strange. There is no control The center line in the graphs of Figure 4e is not required. The order of cell lines in Figure 4e is different from the text. The authors should mention the low cell viability (~80%) in fibroblasts (NHDF-Ad), compared to the other two cell lines (MDA-MB-231 and J774). Anyway, the authors must prove that Mel-NPs are fine for this research purpose. The reviewer cannot understand the contents of “3.2 Intraoperative identification of tattooed lymph nodes”. Also, the reviewer cannot understand what Figures 5b, 5c, 5d, and 5e indicate. The notation of “n=25” is strange (line 300). The abbreviation “MNCs” (line 322) should be removed. The authors should indicate clearly where Mel-NPs are located in Figure 6. Figure 7 should be rebuilt. There is no basis for the ratio of the distribution of Mel-NPs in the tissue. Figure 7 does not indicate the distribution of Mel-NPs. The evaluation on the inflammation by Mel-NPs is meaningless. The reviewer cannot understand why the authors show the size (398 ± 290 µm and 520 ± 223 µm). There is no basis. The explanation on Figure 8 is messed up. The Discussion part should be rebuilt. The reviewer thinks that the authors’ interpretation of experimental results is incorrect (especially, from line 387 to 430). Then, the contents of Conclusions should be changed. The reviewer cannot understand the difference between “the short-term studies” and “the long-term studies” until the last.
Comments on the Quality of English LanguageComprehensive English proofreading should be required.
Author Response
The submitted manuscript describes the development of melanin pigment-containing nanoparticles for the detection of lymph nodes. Although the research object of the submitted manuscript is interesting and the contents and quantity of experiments seem to be sufficient, the authors do not show a sufficient and persuasible explanation for each experiment. The authors should rebuild the last paragraph of Introduction part because it does not reflect the contents of the experimental results. And, the authors should clearly state their opinion for reference [29]. Based on these points, the reviewer judges that the submitted manuscript should be revised largely for publication for “nanomaterials”.
We appreciate Reviewer #2 for the time and critical reading of the manuscript. We very much value the positive insights on the contributions of the manuscript and his/her comments and suggestions. In the revised version of the manuscript we have now restructured the end of the introduction by adding a section where we present recent reported developments using nanomaterials for lymph node labelling. However, to the best of our knowledge, similar approaches for NACT lymph node labelling do not yet exist, which reinforces the interest of this research.
Comment 1: What is PDI? (line 250) The authors should show the basis of “PDI < 0.01”.
Response 1: The PDI value indicates the polydispersity index (i.e., the degree of size dispersion of the particles that make up the sample), considering a PDI value = 0.1 as the upper limit for considering the size distribution of a homogeneous sample (S. Bhattacharjee, "DLS and zeta potential - What they are and what they are not?," J.Control. Release, vol. 235, pp. 337-351, 2016). We have added this reference (48) to clarify and support this concept in line 260 and we have defined the acronym accordingly.
Comment 2: The reviewer cannot understand the meaning of the sentence ”the particle size did not vary substantially from one synthesis to another, although it did increase as the PLGA concentration increased.” (line 246 and 247).
Response 2: We mean that the changes in particle size are very slight, but increase slightly. We have re-phrased this sentence in lines 257-258 for a better understanding ("...particle size increases with PLGA concentration, attributed to an increased viscosity of the organic solution").
Comment 3: Also, the reviewer cannot understand the meaning of the sentence “On the other hands, an attempt was made to obtain colloids with high melanin-load Mel-NPs.”. (line 253)
Response 3: What the authors want to express is that the aim was to increase the concentration of melanin in the particles, since melanin provides the "pigmentation" to the NPs and therefore probably offers a better "tattooing" in the lymph node. To clarify this concept, we have added "as melanin provides the macroscopic colouring necessary for the visualisation of lymph nodes with the naked eye" in lines 264-265.
Comment 4: The authors should describe the intention of showing the pH of Milli-Q water.
Response 4: As the pH of the water increases, there is an increase in the ionisation of the functional groups present in melanin due to de-protonation (COOH and NH2). Ionisation of these groups increases the negative charge of the melanin molecule, which facilitates supramolecular interactions with water molecules. These interactions allow the melanin molecules to disperse more efficiently in water, as the additional negative charges repel the melanin molecules from each other, preventing aggregation and allowing more uniform dispersion. To emphasise the need for this, we have added "Increasing the pH of the water to pH 8 provides an environment in which melanin can be properly ionised, increasing its solubility and allowing a better dispersion in water" on lines 268-270.
Comment 5: The reader cannot see the scale bar in Figure 3a.
Response 5: In agreement with the reviewer, we have updated the scales in Figure 3. Thank you for the concern.
Comment 6: The authors should explain why the particle size of Mel-NPs decreases in the presence of melanin.
Response 6: The reviewer has raised an interesting question. Firstly, the incorporation of melanin within the PLGA matrix can alter the configuration and packing of the polymer chains. When a molecule is encapsulated, it can induce a rearrangement of the PLGA chains, resulting in a more compact structure and thus a decrease in nanoparticle size. Additionally, the presence of the encapsulated molecule can influence the viscosity and surface tension of the forming nanoparticle. During the emulsification and solvent evaporation process, the encapsulated molecule can act as a stabilizing agent, decreasing the interfacial energy between the aqueous and organic phases. This can lead to the formation of smaller and more stable nanoparticles. This observation was also cited by other authors (i.e. https://doi.org/10.1016/j.ajps.2015.09.004).
Comment 7: While Figure 3a and Figure 3b show clear SEM images of synthesized particles, Figure 4a shows the indecisive TEM image of them. If the synthesized Mel-NPs are not so fragile, the authors should show the robustness of them.
Response 7: A new TEM image is proposed in the revised version of the manuscript where it can be observed a low magnification image to convince the reviewers that the nanoparticles are homogenous and an inset including a high magnification image where a PLGA nanoparticle and inner melanin nanodeposits (as dark shadows inside) are depicted.

Comment 8. It is difficult to evaluate the amounts of melanin encapsulated in Mel-NPs by the TG curve shown in Figure 4b. Since Figure 4c just indicates the UV spectrum of Mel-NPs, Figure 4c does not indicate the amount of melanin encapsulated in Mel-NPs. The authors must show the UV spectrum of PGLA particles. Based on these points, the reviewer thinks that the amounts of melanin encapsulated in Mel-NPs cannot be estimated from Figure 4b and Figure 4c.
Response 8: We evaluated indirectly the amount of melanin encapsulated by measuring the supernatant which includes free melanin only (without PLGA). Therefore, we did not consider it necessary to add the UV-Vis curve of the Mel-NPs for this purpose.
Comment 9: Additionally, the reviewer cannot understand the meaning of the sentence “a melanin encapsulation yield of 65.8 ± 9.1 %, which represents 23.5 % loading by weight in the particles.”. (line 278 and 279)
Response 9: We are sorry if this section is not well understood. The encapsulation efficiency is calculated from the melanin that is initially introduced during the synthesis. So, for example, if we start with 100 ug of melanin, we only manage to encapsulate 65.8 ug, that represents an encapsulation efficiency of 65.8%. However, the value of 23.5% refers to the amount of melanin in the particles after synthesis, which is generally referred as drug loading. This value is obtained after dividing the amount of entrapped melanin in the particles divided by the total amount of melanin-containing nanoparticles. We have rewritten this sentence for a better understanding ("With this method, a melanin encapsulation yield of 65.8 ± 9.1 % (with respect to the initial concentration used in the synthesis), which represents 23.5 % loading by weight in the resulting Mel-NPs.") in lines 292-294.
Comment 10: The sentence “For the experiment, Mel-NPs were dispersed in PBS 1x and kept for 28 days at 37°C under continuous agitation.” (line 281 and 282) is not clear. And, the sentences from line 282 to 287 suggest the synthesized Mel-NPs are fragile in PBS. If so, the reviewer thinks that the synthesized Mel-NPs will not be available for any bioassay.
Response 10: We apologize for any misunderstanding that that sentence might have caused. The particles are stable in PBS over several weeks, even months, and, of course, during the biological assays, without aggregation or precipitation (doi: 10.1016/j.jconrel.2012.01.043). What we intended to explain in the text is that within the duration of the experiment, a sustained release of a small percentage of the encapsulated melanin is observed by the diffusion from the most superficial part of the particle to the external medium.
PLGA is a biodegradable polymer that in contact with water gets hydrolyzed by breaking its ester bond and releasing the composing monomers lactic acid and glycolic acid. However, the biodegradation rate is slow (the polymer we used reaches a complete degradation in 3 months) Such a slow degradation rate is used to provide with a sustained release of the encapsulated melanin and to last in the injected lymph node for extended periods of time. Initially only a small amount would be released. So the particles are stable in PBS and only the melanin present in the outermost layer of the polymeric matrix gets released at the beginning. We used that polymer to complete degradation and avoid any bioaccumulation after tattooing.
Comment 11: The description on the cytotoxicity of the synthesized Mel-NPs using cell lines is strange. There is no control The center line in the graphs of Figure 4e is not required.
Response 11: As explained in the materials and methods section (lines 166-168), cytotoxicity was assessed by comparing the values obtained for treated cells with those recovered from untreated cells, assigning 100% viability to these controls. To clarify this method for the reader, we have added " Cytotoxicity is determined by assigning 100% cell viability to untreated control cells " to line 312 of the results section. We have not incorporated PLGA nanoparticle controls (without encapsulated melanin) because in this work we are interested in studying the cytotoxic effect of Mel-NPs, not free melanin. We have also removed the line from the graphs, as it only expressed 70% cell viability in order to consider the nanomaterials as non-cytotoxic as the ISO 10993-5 standard norm for biomedical devices states.
Comment 12: The order of cell lines in Figure 4e is different from the text. The authors should mention the low cell viability (~80%) in fibroblasts (NHDF-Ad), compared to the other two cell lines (MDA-MB-231 and J774). Anyway, the authors must prove that Mel-NPs are fine for this research purpose.
Response 12: We have corrected the order of the cell lines in the text (lines 303-305). According to ISO 10993-5, a compound or material is not considered cytotoxic if it does not reduce cell viability below 70% compared to the untreated control cells (assigned with 100% viability). In the results obtained for the fibroblast cell line, none of the tested concentrations reduced the viability below 80%, indicating that Mel-NPs are not cytotoxic at the concentrations tested for the purposes of this research. To clarify the results, the sentence “The results are in compliance with ISO 10993-5 [Ref: https://www.iso.org/standard/36406.html], which states that cell viability higher than 70% is considered as cytocompatibility” has been added on the lines 307-309.
Comment 13: The reviewer cannot understand the contents of “3.2 Intraoperative identification of tattooed lymph nodes”. Also, the reviewer cannot understand what Figures 5b, 5c, 5d, and 5e indicate.
Response 13: The identification rate is the percentage of nodes that have been identified during surgery. That is, they have served the purpose of tattooing the lymph node in the long term. On the other hand, "ease of identification" refers to the persistence of the tattoo in the lymph node. In other words, the more tattoo remains in the lymph node (and therefore the blacker it is), the easier it would be to identify it at surgery. This is explained in more detail and with examples in Figure 2 of materials and methods. For a better understanding of this section, we have added lines 325-326: "As depicted in Figure 5b, 77.8-83.3% of the tattooed nodes were identified during the surgery (identification rate)"; lines 328-329 ("Also, in both, short-term and long-term studies, the nodes showed clearly visible tattoos (Figure 5c)").
Comment 14: The notation of “n=25” is strange (line 300).
Response 14: We have replaced “n=25” (i.e., the sample size of 25) with "a total of 25" to improve comprehension. We believe that this section is now better understood.
Comment 15: The abbreviation “MNCs” (line 322) should be removed. The authors should indicate clearly where Mel-NPs are located in Figure 6.
Response 15: We agree with the reviewer. We have removed the abbreviation and added arrows in Figure 6 for an easier identification. Additionally, we added “where Mel-NPs appear brown and are easily distinguished from the H&E staining” to clarify the location of Mel-NPs in lines 350-351.
Comment 16: Figure 7 should be rebuilt. There is no basis for the ratio of the distribution of Mel-NPs in the tissue. Figure 7 does not indicate the distribution of Mel-NPs.
Response 16: We have studied the localisation of the particles as part of his/her in vivo characterisation. The localisation of Mel-NPs might be expected to affect inflammation in the lymph node. For example, there are more immune cells in the inner and pericapsular regions that could be altered/activated as part of the foreign body response mechanisms. However, as expected, NPs that are mostly found in adjacent tissue (mainly collagen and fibroblasts) have less impact on lymphatic tissue. In figure 7 we have incorporated examples we have found of the various locations in the lymph node. However, we have not found any characteristics that bias the NPs towards one location or another, which could lead to future studies to better understand these aspects. We consider this aspect to be of great importance in lymph node tattooing. Since we consider this important, we believe it is necessary to keep it in the manuscript to provide this information to other authors and or future research work.
Comment 17: The evaluation on the inflammation by Mel-NPs is meaningless. The reviewer cannot understand why the authors show the size (398 ± 290 µm and 520 ± 223 µm). There is no basis.
Response 17: Foreign body-type inflammation occurs in all tattooed lymph nodes, characterised by the presence of activated macrophages that phagocytose the NPs, as well as macrophages that fuse into multinucleated giant cells to isolate the NPs. Quantification of inflammation beyond the presence/absence of these events is meaningless. However, we have considered that quantifying the extent of inflammatory injury (characterised by tissue structural change) makes sense. We have selected on all cases the same area and quantified the extension of the inflammation. Of course, we have considered other markers of tissue damage, such as necrotic foci. However, we believe that measuring the extent of the inflammatory reaction is a parameter that allows us to assess differences in injury between nodes.
Comment 18: The explanation on Figure 8 is messed up.
Response 18: We have added some additional explanations in the caption of Figure 8 to improve its understanding. We believe that now it is better understood.
Comment 19: The Discussion part should be rebuilt. The reviewer thinks that the authors’ interpretation of experimental results is incorrect (especially, from line 387 to 430). Then, the contents of Conclusions should be changed.
Response 19: We humbly believe that the discussion part has a logic structure. First we describe previous reported clinical trials where SLNs labeling studies have been carried out. We discuss advantages and disadvantages of the different approaches followed. We discuss the advantages of our proposed strategy. And we summarize the in vitro and in vivo results obtained. We highlighted the successful in vivo identification of the labeled nodes and we proposed future work in the field. We think that this logic structure is easy to follow and helps others working in the field.
Comment 20: The reviewer cannot understand the difference between “the short-term studies” and “the long-term studies” until the last.
Response 20: We have added "The short-term studies consisted of stabling for 1-6 weeks, while the long-term studies were conducted 16 weeks after the administration " in line 321-323 to remind the reader of the meaning of the short-term and long-term studies in the first part of the results.
Thank you again.

Round 2
Reviewer 1 Report
Comments and Suggestions for Authors
This manuscript has improved significantly by the authors, and all my questions have been well-resolved.
Author Response
Comment 1: This manuscript has improved significantly by the authors, and all my questions have been well-resolved.
Response 1: We thank reviewer 1 for his/her comments and his positive assessment of our work.
Reviewer 2 Report
Comments and Suggestions for Authors
The reviewer thinks that the authors do not show any basis for all numbers (ratio and size) that the authors present in the submitted revised manuscript. The reviewer cannot read the numbers (e.g. 50%, 35.7%, and 14.3% in line 361; 264 nm and 43 nm in line 383) from the corresponding Figures. As long as the authors do not explain them clearly, the reviewer judges that the submitted revised manuscript is “reject”.
Comments on the Quality of English LanguageModerate English proofreading is required.
Author Response
Comment 1: The reviewer thinks that the authors do not show any basis for all numbers (ratio and size) that the authors present in the submitted revised manuscript. The reviewer cannot read the numbers (e.g. 50%, 35.7%, and 14.3% in line 361; 264 nm and 43 nm in line 383) from the corresponding Figures. As long as the authors do not explain them clearly, the reviewer judges that the submitted revised manuscript is “reject”.
Response 1: We thank the reviewer for his/her critical review of our work. In agreement with him/her, we have generated a "Supplementary" file to include all data used that are not within the main manuscript. The reviewer is totally right, that data was included in the main body of the manuscript but no the original data which generate those values. Now we have included all the original data. We believe that our article has now been significantly improved.
We have double-checked the English style and believe it is now correct. We would like to thank the reviewer again for his/her contributions to our work.